# Consistency of Yield Ranking and Adaptability Patterns of Winter Wheat Cultivars between Multi-Environmental Trials and Farmer Surveys

**Marcin Studnicki** [1,*], **Manjit S. Kang** [2], **Marzena Iwańska** [1], **Tadeusz Oleksiak** [3], **Elżbieta Wójcik-Gront** [1] **and Wiesław Mądry** [1]

1   Department of Experimental Design and Bioinformatics, Warsaw University of Life Sciences, Nowoursynowska, 15902-766 Warsaw, Poland; marzena_iwanska@sggw.pl (M.I.); ewojcik.gront@gmail.com (E.W.-G.); wieslaw_madry@sggw.pl (W.M.)
2   Department of Plant Pathology, Kansas State University, Manhattan, KS 66506-5502, USA; manjit5264@yahoo.com
3   Plant Breeding and Acclimatization Institute–National Research Institute (IHAR-PIB), Radzików, 05-870 Błonie, Poland; t.oleksiak@ihar.edu.pl
*   Correspondence: marcin_studnicki@sggw.pl; Tel.: +48-22-59-32-727

**Abstract:** Cultivar recommendation based on mean performance determined by multi-environment trials (METs) conducted on research stations could be unreliable and ineffective for assessing performance in farmers' fields. It is important to improve the efficiency of cultivar recommendation based on METs. For this purpose, it would be useful to validate recommendations based on yield data obtained directly from farmers, i.e., through surveys. The aim of this study was to discuss the possibility and statistical methodology of assessing cultivar performance patterns based on yield data obtained through farmer surveys. We suggest that this might be accomplished by assessing the conformity of yield ranking and yield performance patterns between MET and survey datasets in the same growing regions. As an example, we compare winter wheat (*Triticum aestivum* L.) yield data obtained from Polish farmers via surveys with data obtained via METs. In the METs, cultivars were evaluated at two levels of crop-management, a moderate-input management (MIM) system and a high-input management (HIM) system. Based on the yield evaluations in the current study, half of the agro-ecological regions had relatively high levels of consistency in yield rankings between the MET MIM system and survey yield dataset. This indicated a relatively high efficiency of cultivar recommendations based on METs in these regions, especially for the MIM system. For the HIM system, however, with the exception of one region, we observed a poor degree of consistency in cultivar ranking.

**Keywords:** crop management; cultivar recommendation; genotype × environment interaction; mixed linear model

## 1. Introduction

The main goal of many breeding programs is to obtain cultivars with broad adaptation to as many environments as possible. Such cultivars are labeled as widely adapted and they allow breeding companies to achieve market and financial success. Wide adaptation is defined as the ability to produce relatively high yields consistently across diverse agricultural environments in a growing region (spatial stability) [1,2]. On the other hand, breeders and farmers specifically look to narrowly adapted cultivars [3] for specific production systems or conditions, such as organic agriculture. Growers would prefer cultivars that perform consistently at their location year after year (temporal stability).

Cultivars are recommended based on an assessment of their performance. It is also important to assess cultivar stability according to the dynamic concept [4]. A stable cultivar is one that has yield parallel to the mean yield of all cultivars (environmental means) across environments. Adaptation patterns of cultivars and their stability are related to the extent of genotype × environment (G × E) interaction. The most stable cultivars are characterized by a negligible G × E interaction effect. Usually, the evaluation of G × E interaction and adaptability patterns of cultivars is carried out through controlled METs [2,5].

METs are experiments where selected cultivars are evaluated in multiple locations, across several years, and sometimes at two or more crop-management (intensity) levels. Such experiments are carried out at highly specialized experimental stations that are quite often part of universities, cultivar registration offices, breeding companies, and other institutions (e.g., agricultural advisory services). Those experimental stations employ highly qualified staff and experiments are carried out at a high level of crop management, adhering to the principles of good agricultural practices. Quite often, such degrees of advancement in crop cultivation are not available to average farmers, who obtain lower yields on their farms. For example, the average yield of winter wheat in the Polish MET system (Post-Registration Variety Testing System (PVTS)) used to recommend cultivars is >8 tones ha$^{-1}$, whereas the average yield obtained by farmers is only around 4.5 tones ha$^{-1}$ [6].

The difference between potential yield, i.e., obtained in official experiments (controlled trials), like PVTS, and the yield obtained by farmers (actual yield) is referred to as the yield gap [7,8]. Reduction in the yield gap can significantly contribute towards increasing crop productivity [9], resulting in enhanced production without increasing the area under cultivation. To meet the expected food needs of the increasing world population and for biofuels, crop production needs to be doubled by 2050 [6,10].

Low yields might result from the use of cultivars that are not adapted to farmers' production systems and environmental conditions existing at their farms. Consequently, cultivar recommendations based on adaptation patterns estimated in the official METs may be unreliable under actual farm conditions. Thus, it is important to improve the efficiency of cultivar recommendations based on METs. For this purpose, it may be useful to validate the MET recommendations using yield data obtained directly from farmers, e.g., in the form of surveys. It might become the basis to suggest changes in cultivar recommendation based on METs.

In many countries, researchers and survey-makers have been collecting data on crop management and crop yields achieved by farmers for many years. An example of such research is surveys conducted by the U.S. Department of Agriculture (USDA) and reported, inter alia, in "Crop production" reports. In Europe, "Farm Practices Surveys" are carried out by the United Kingdom government, or in France, surveys are carried out by the L'Office National Interprofessionnel des Grandes Cultures ONIGC and Arvalis-Institut du Végétal [11]. Such surveys are a valuable source of information on farmers' agricultural production systems in a given region/country. Farmer survey data have been commonly used to evaluate the effectiveness of cultivar recommendations, plant protection and fertilization systems. Such surveys are also used to assess the impact of crop management and soil tillage on crop yields. However, this type of data has not been used to assess the stability or/and adaptability of cultivars and to validate the cultivar recommendations derived from METs. The use of survey data is, however, associated with many problems and difficulties in assessing G × E interaction and adaptability patterns of cultivars. The main reason for this is the variable number of repetitions for the tested varieties in growing regions.

The aim of this study was to present a statistical methodology for assessing cultivar adaptability patterns based on yield data obtained from farmer surveys. We propose approaches for validation of cultivar evaluation and make an attempt to evaluate the effectiveness and efficiency of cultivar recommendations based on METs. We suggest that it might be accomplished by assessing the conformity of yield ranking and yield adaptability patterns between the METs and survey datasets within a growing region. As an example, we used winter wheat yield data from Polish farms obtained via surveys and METs, i.e., from the Polish Post-Registration Variety Testing System.

## 2. Materials and Methods

### 2.1. Yield Datasets

We used two datasets representing winter wheat yields of 12 cultivars, obtained from eight growing seasons, i.e., 2008 to 2016 (Table 1). The first dataset included grain yield data from 49 locations (Figure 1) of the Polish Post-Registration Variety Testing System (PVTS); we will call it the MET dataset. Each of the 49 trial locations was assigned to one of six agro-ecological regions [12]. Cereal growing areas in Poland have been divided into six agro-ecological regions (Figure 1). The division into regions roughly reflects soil and climatic conditions in Poland. In each trial, the winter wheat cultivars were evaluated at two levels of crop-management intensity, a moderate-input management (MIM) system and a high-input management (HIM) system. The MIM system used a relatively low rate of nitrogen fertilization (approximately 100 kg N per hectare; 40 kg of N at BBCH 29 of phenological development stages scale proposed by Reference [13] and the remainder at BBCH 49). No fungicide was used to control leaf diseases. The second level, HIM, was an intensive wheat production system with an additional fertilization of 40 kg of N per ha and fungicide applications at BBCH 31–32 and BBCH 49–60 to protect against diseases as well as spraying at BBCH 31 to reduce lodging. Fungicidal active agents were selected and used, depending on the severity of fungal diseases. Each field experiment was conducted according to a two-factor (crop management and cultivar) strip-plot design with two replications, using a resolvable incomplete block design for cultivars. The area of each plot was 15 m$^2$. The yield observations in the MET dataset were unbalanced mainly because of the different number of trial locations in particular growing seasons.

**Table 1.** Characteristics of the cultivars.

| Cultivar Name | Country of Origin | Cultivar Type [a] |
|---|---|---|
| Akteur | Germany | A |
| Bamberka | Poland | A |
| Batuta | Poland | B |
| Bogatka | Poland | B |
| Finezja | Poland | A |
| Jenga | Germany | B |
| KWS Ozon | Germany | B |
| Legenda | Poland | A |
| Ludwig | Austria | A |
| Ostroga | Poland | A |
| Smuga | Poland | A |
| Tonacja | Poland | A |

[a] Polish quality scheme (A: good quality cultivar, B: bread cultivar).

The second dataset represented surveys performed as part of a study conducted by the Plant Breeding and Acclimatization Institute (IHAR)—National Research Institute, located in Radzików, in cooperation with Regional Agricultural Advisory Centers. This dataset will be designated as the survey dataset. The questionnaire sought information about cultivars and cultural practices used, e.g., fertilization, plant protection measures, crop rotation, and tillage systems for winter wheat cultivation. These data were collected from about 400 farms by agricultural advisors in each growing season. The farms were randomly selected across Poland from among those that used services of Regional Agricultural Advisory Centers. Similar to the trial locations for the MET dataset, the surveyed farms were divided into six agro-ecological regions (Figure 1). The datasets from farm surveys, as from METs, were highly unbalanced as a result of the variable number of surveyed farms in a given growing season and in a given agro-ecological region. In addition, the survey dataset involved different sets of cultivars grown on surveyed farms in each region.

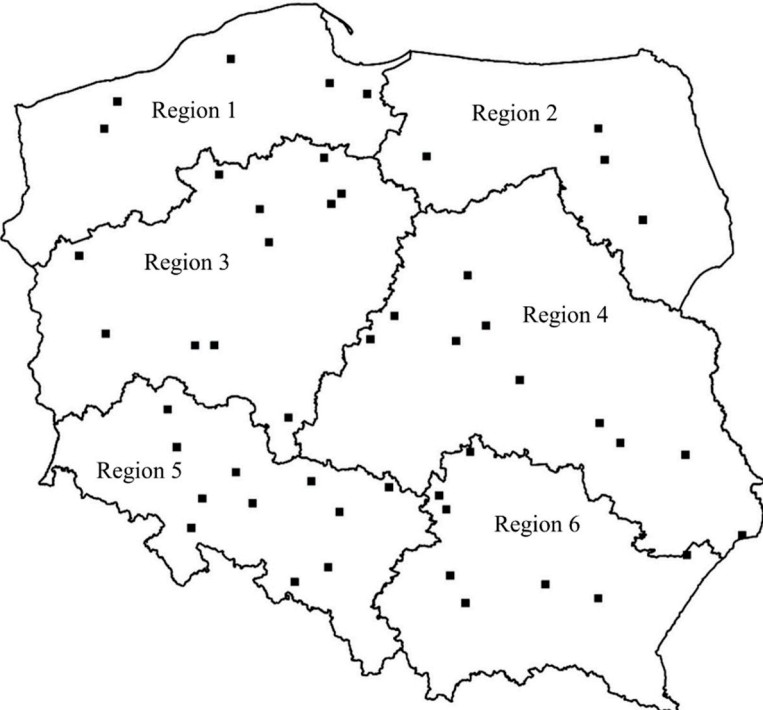

**Figure 1.** The 49 locations used to generate multi-environmental trial datasets across the six agro-ecological regions in Poland.

## 2.2. Statistical Analysis

Due to the large imbalance in both (MET and survey) yield datasets, our data analysis took a single-stage approach using a linear mixed model (LMM) with restricted maximum likelihood (REML) to estimate model parameters. In more complex METs, a cultivar recommendation is usually given for each of the crop-management intensities separately. LMMs used for statistical analysis of complex METs do not commonly contain crop management factors [14], as they increase model complexity. Therefore, for the MET dataset, yield observations were analyzed separately for MIM and HIM systems. The linear mixed model used for this dataset was as follows:

$$
\begin{aligned}
y_{ijhklmq} = {} & \mu + z_q + zl_{q(j)} + g_k + a_i + za_{qi} + gz_{kq} + glz_{kq(i)} + ga_{ki} + zla_{iq(j)} + \\
& gza_{kqi} + gzla_{kq(j)i} + r_{jih} + b_{jihm} + e_{ijhklmq}
\end{aligned}
\tag{1}
$$

where $y_{ijhklmq}$ is the winter wheat yield for one crop management level; $\mu$ is the overall mean; $z_q$ is the fixed effect of the $q$th agro-ecological region; $g_k$ is the random effect of the $k$th cultivar; $a_i$ is the random effect of the $i$th year; $zl_{q(j)}$ is the fixed effect of the $j$th location nested within the $q$th agro-ecological region; $za_{qi}$ is the random interaction effect of the $i$th year and the $q$th agro-ecological region; $gz_{kq}$ is the random interaction effect of the $k$th cultivar and the $q$th agro-ecological region; $glz_{kq(j)}$ is the random interaction effect of the $k$th cultivar and the $j$th location nested in the $q$th agro-ecological region; $ga_{ki}$ is the random interaction effect of the $k$th cultivar and the $i$th year; $zla_{iq(j)}$ is the random interaction effect of the $i$th year and the $j$th location nested in the $q$th agro-ecological region; $gza_{kqi}$ is the random interaction effect of the $k$th cultivar, $q$th agro-ecological region and $i$th year; $gzla_{kq(j)i}$ is the random interaction effect of the $k$th cultivar, $i$th year and $j$th location nested in the $q$th agro-ecological region; $r_{jih}$ is the random effect of the $h$th replication nested in the $j$th location at the $i$th year; $b_{jihm}$ is the random effect of the $m$th block nested in the $h$th replication at the $j$th location and the $i$th year; and $e_{ijhklmq}$ is the random error associated with the yield observation $y_{ijhklmq}$.

For the yield observations from farmer surveys (survey dataset), the following LMM was used:

$$x_{kiq} = \mu + z_q + g_k + a_i + za_{qi} + gz_{kq} + ga_{ki} + gza_{kqi} + e_{kiq} \tag{2}$$

where $x_{kiq}$ is wheat yield from a studied farm; $\mu$ is the overall mean; $z_q$ is the fixed effect of the $q$th agro-ecological region; $g_k$ is the random effect of the $k$th cultivar; $a_i$ is the random effect of the $i$th year; $za_{qi}$ is the random interaction effect of the $i$th year and the $q$th agro-ecological region; $gz_{kq}$ is the random interaction effect of the $k$th cultivar and the $q$th agro-ecological region; $ga_{ki}$ is the random interaction effect of the $k$th cultivar and the $i$th year; $gza_{kqi}$ is the random interaction effect of the $k$th cultivar, $q$th agro-ecological region and the $i$th year and $e_{kiq}$ is the residual error.

For this linear mixed model, individual farms in a given region and growing season were treated as replications. The number of surveyed farms (i.e., replications) in individual regions and growing seasons varied. For the random effects of cultivars in each agro-ecological region in models (1) and (2), we assumed a variance-covariance matrix where the diagonal elements represented cultivar variances in a region and the off-diagonal elements represented cultivar covariances between pairs of regions [15]. That structure was modeled by the factor analytic structure with four components—FA (4). The FA structure uses a multiplicative approximation of the unstructured variance-covariance matrix. In many other studies, factor analytic structures were recommended in modeling G × E (e.g., location, region) interactions for unbalanced data [16,17]. The linear mixed models (1) and (2) for both datasets were used to obtain adjusted yield means for the cultivar-agro-ecological region combinations across studied growing seasons. However, for the MET dataset, the adjusted means for a cultivar-agro-ecological region were calculated separately for the MIM and HIM systems. These means were calculated using the algorithm described by Welham et al. [18] and was used in further stages of this study. The consistency of the cultivar yield rankings across regions obtained on the basis of adjusted means between survey and MET datasets were evaluated using Spearman's rank correlation coefficients ($r_s$).

The adjusted means for the cultivar-agro-ecological region combinations were estimated with different prediction accuracies. Therefore, for reliable evaluation of cultivar adaptability, the adjusted means for cultivar-agro-ecological region interactions were weighted using the standard error of the mean (SEM). The SEM was separately calculated for the adjusted cultivar yield means for each region based on published methods [18,19].

Using the adjusted means for cultivar-agro-ecological regions, we described cultivar adaptation to each environment (region) with the help of GGE (genotype main effects plus G × E interaction effects) biplot analysis [20]. The biplot analysis is based on the environment-centered (in our case region-centered) principal component analysis (PCA) [21,22]. Due to the different prediction accuracies of the cultivar-agro-ecological region-adjusted means to prepare GGE biplot, we used the weighted expectation maximization PCA (wEMPCA) approach [23]. This approach is identical to the weighted low-rank, singular value decomposition (SVD) method proposed by Rodrigues et al. [24] for use in the additive main effects and multiplicative interaction (AMMI) model. In wEMPCA, for a more accurate estimation of adjusted means, the weights should be smaller when adjusted means are estimated with larger errors. For the wEMPCA approach, we computed and used the parameter $w_{ik}$ as a weight for the adjusted means; the formula for $w_{ik}$ is as follows:

$$w_{ik} = \frac{1}{SEM_{ik}^2} \tag{3}$$

where $w_{ik}$ is the weight for the adjusted mean of the $i$th cultivar in the $k$th region, and $SEM_{ik}$ is the standard error of the adjusted mean of the $i$th cultivar in the $k$th region.

For the statistical analysis, we used the R 3.2.5 software package. The linear mixed models from equation (1) and (2) were fitted using ASReml 3.0, implemented in the R software package ASReml-R [19]. The wEMPCA method used the code proposed by Bailey [23], which is available at https://github.com/sbailey/empca/.

## 3. Results

For the MIM system, there were varying degrees of consistency of cultivar yield rankings between the survey dataset and the MET dataset in different growing regions (Table 2). The best agreement of cultivar yield ranking between survey and MET datasets was for regions 2, 3 and 6. In these regions, the Spearman's rank correlation coefficients ($r_s$) were positive and >0.60. For the other three regions, the correlation coefficients were low ($r_s \leq 0.401$), indicating a lack of consistency of cultivar yield ranking between the two datasets.

**Table 2.** Correlation coefficients ($r_s$) for cultivar yield ranking between the survey dataset and the multi-environmental trial (MET) dataset.

| Agro-Ecological Region | Crop Management for MET Experiments | |
| --- | --- | --- |
| | Moderate-Input | High-Input |
| Region 1 | 0.295 | 0.301 * |
| Region 2 | 0.632 ** | 0.205 |
| Region 3 | 0.662 ** | 0.199 |
| Region 4 | 0.401 * | 0.231 |
| Region 5 | 0.191 | 0.202 |
| Region 6 | 0.704 ** | 0.598 ** |

*,** Significant at the 0.05 and 0.01 probability level, respectively.

The cultivar yield rankings between the survey dataset and the HIM system (MET) dataset in different regions were generally characterized by poor agreement (Table 2). In only region 6, a significant association between the two datasets was observed ($r_s \approx 0.60$).

Cultivars widely adapted to specific regions were identified on the basis of the GGE biplots (Figure 2). KWS Ozon was a widely adapted cultivar in all studied datasets for most regions, whereas Legenda was widely adapted to regions only in the survey dataset and for the MIM system. Regardless of the dataset, the variable region was characterized by low discriminative power regarding cultivars, possibly indicating a relatively weak significance of G × E interaction. G × E interaction was slightly larger for the MIM and HIM systems for the MET dataset than for the survey dataset.

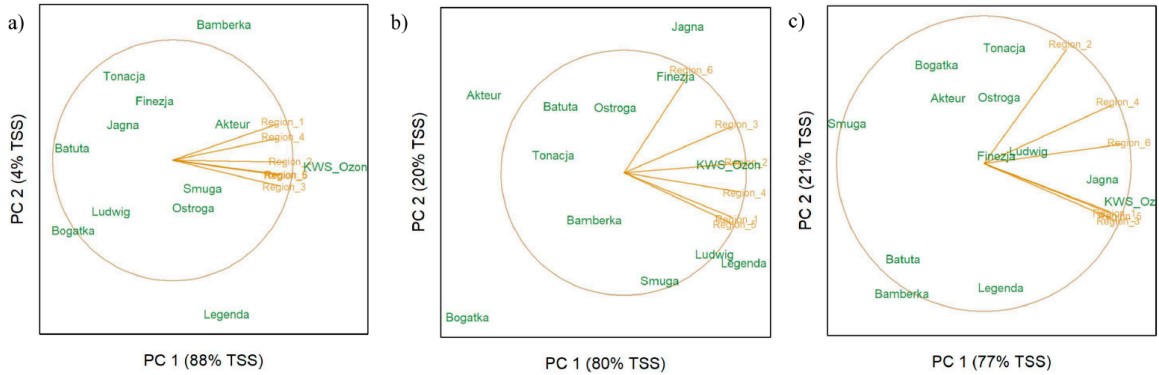

**Figure 2.** GGE (genotype main effects plus genotype × environment interaction effects) biplots based on adjusted means of cultivar × region combinations for the survey dataset (**a**), for the moderate-input crop management (MIM), multi-environmental trial dataset (**b**), and for the high-input crop management (HIM), multi-environmental trial dataset (**c**). PC, principal component; TSS, total sum of squares.

## 4. Discussion

To date, the assessment of the adaptability of cultivars for different crops has been carried out using only METs. To supplement this information and to improve on-farm yields, yield data obtained directly from individual farms via farmer surveys are needed. In this paper, we have proposed and

performed a statistical methodology that allows assessment of the adaptability of crop cultivars based on data from farmer surveys. On the basis of proposed methods, we compared the consistency of yield rankings of cultivars between MET and farmer survey datasets. Such information can be used for validation and assessment of the effectiveness of cultivar recommendations based on METs.

Based on the yields of winter wheat cultivars evaluated in the current study, half of the agro-ecological regions were found to have a relatively high level of consistency in yield rankings between the MIM system (MET) and the survey dataset. This indicated a relatively high level of effectiveness of cultivar recommendations based on METs in these regions for the MIM system. However, in the other half of regions for the MIM system, this consistency was much weaker, indicating that the recommendations based on the MIM system were not highly effective and needed to be improved. The main reason for this result could be a poor choice of trial locations, which were not representative of the climatic and soil conditions of a given region. Remedial action could be to replace test locations with more appropriate ones or to increase the number of locations in such regions. Of course, there could be other reasons for the lack of agreement between MET and survey datasets from these regions. Unfortunately, it was difficult to identify other causes from our datasets. Additional studies will be needed to clearly identify the reasons for this situation. Perhaps, use of soil properties as covariates to remove heterogeneity from G × E interaction, as shown by Kang and Gorman [25] in maize and in van Euwijk et al. [26], could shed light on the possible causes of inconsistency.

Regions with low, or lack of, consistency in cultivar yield rankings included those located in the western part of the country (regions 1 and 5), which were characterized by warmer climatic conditions, and in the eastern part (region 4) with much lower average temperatures and a shorter growing season. However, for region 4, the Spearman rank correlation coefficient was twice as high as those for regions 1 and 5. In addition, we assumed that the availability of water was irrelevant. Both regions with high cultivar yield ranking compliance and among regions with low compliance, there were locations that were characterized by high and low water availability. For example, Region 3 with high consistency of cultivar yield ranking between the survey and MET datasets was characterized by the largest water deficit among all wheat-growing regions in Poland. Agro-ecological regions with low consistency in cultivar yield between MIM system (MET) dataset and survey dataset were characterized by a high diversity of farm types. These regions also had the largest proportion of relatively large farms (often >1000 ha).

For the HIM system, apart from one region, there was a low degree of consistency in winter wheat yields. The HIM system, with very high inputs, is very rarely found on farms in Poland. According to Wójcik-Gront [27], the additional yield from the HIM level in relation to the MIM level was not sufficiently large to compensate for the extra expenditure. The comparison of MET and survey datasets indicates that recommendations based on HIM may not be effective. Only Region 6 with unfavorable environmental conditions for winter wheat had consistency in cultivar yield rankings between MET and survey datasets. The soil quality in this region was the poorest among all growing regions and varietal performance obviously responded to the increased inputs. A recent study on winter wheat in Poland [12] showed a large consistency in cultivar rankings between the MIM and HIM systems in this region.

The inconsistency of the cultivar yield described in the above paragraphs does not necessarily indicate an erroneous or ineffective cultivar recommendation based on METs. It can only signal that such a recommendation is problematic and requires special attention of scientists. Such inconsistency may indicate poor or incomplete division of the growing area into agro-ecological regions or even poor selection of trial locations that represent a given region. Unfortunately, the method of data collection and anonymous surveys did not allow us to propose a better division of the growing area into agro-ecological regions. Survey data could support the decision to develop more effective agro-ecological regions. Many previous studies based on METs have indicated that cultivar yield adaptability patterns differ depending on crop management [28–30]. Often, surveys contain data from highly diverse crop management systems, such as organic, integrated, and high input, all of which

can affect cultivar adaptation patterns. Improved data can be obtained by separately assessing the adaptability of cultivars for groups of farms with similar crop management systems.

Unfortunately, survey yield data are not ideal and cannot always be used to reliably evaluate the effectiveness and validate cultivar adaptability. In particular, a reliable and effective statistical analysis of such data is difficult. One of the basic disadvantages and difficulties of using survey data is the lack of homogeneous crop management across sampled farms, as virtually every farm has its own crop management. It would be worth considering the use of trials for cultivar recommendation on farms rather than experiment stations. In Florida, for example, the last two stages of sugarcane variety trials are conducted on farms; those replicated trials are managed by farmer cooperators but data are collected by scientists. In survey datasets, this factor is not strictly controlled, unlike in the METs. However, the surveyed farms (fields) were selected randomly and can be considered a representative sample of the population of crop managements in each region. The low consistency of cultivar yield rankings in the HIM system confirms this. HIM system represents very luxurious crop management, which is rarely found at the practical farm level in Poland.

Another difficulty in applying the proposed methods, especially those based on adjusted means, i.e., GGE, is the calculation of means with equal precision. Across regions, the number of surveys and the number of cultivars used by farmers varied. That is why we proposed weighting of adjusted means. A similar procedure is used for the two-stage analysis using observations from METs. Möhring and Piepho [31] suggested that results from the analysis of METs without weighted means had an acceptable level of reliability. However, their results related to controlled experiments with the same number of replications per cultivar. Unfortunately, for yield data obtained through surveys, this condition cannot be met. To ensure a high level of credibility of results from this type of data, it is necessary to apply the weighting in further stages of cultivar adaptability evaluation. The literature describes a number of methods for weighting data from METs based on various statistics in multistage analyses. However, the SEM-based method may prove to be the most effective. This parameter not only takes into account the variability but also the number of observations. The major advantage of this parameter relates to the fact that in many statistical applications, it is relatively easy to access, especially the tools used in the classical estimation of linear mixed models, such as SAS, ASReml or GENSTAT. This relatively simple procedure of weighting of adjusted means from the proposed model will contribute to a more credible conclusion from evaluations of cultivars coming from survey studies.

On the other hand, the farmer surveys are characterized by a low level of credibility. They rely only on information obtained from farmers, which is often not precise or accurate. Frequently, farmers do not assess the yield value exactly, although today's technique allows it; for example, by using harvesters that allow to us develop field yield maps. Survey data should meet certain requirements to use classical statistical methods to assess the adaptability of cultivars. The data should constitute a representative sample of farms in a given area or region. Otherwise, it is necessary to weight the adjusted means used for analysis and cultivar evaluation. Unless the frequency of cultivars is the same, weighting is necessary to improve the reliability of results. Assessment of adaptability based on the use of data obtained directly from farms may contribute to a better assessment of the yield gap and its variation across regions, especially when surveys contain detailed information about crop management used on studied farms. Quite often, such surveys are carried out for many years (e.g., data from Polish farms have been collected since 1998). The data could be used to assess farming practices, which could help in selecting climate-resilient cultivars [32,33].

A few studies on evaluation of cultivar adaptation have been carried out separately for each year or growing season [34,35]. These evaluations and recommendations are hardly ever credible and hence might not be very useful for growers. Given the year-to-year variation in climatic conditions, cultivar evaluations must be based on data from several years using yield-adjusted means. These approaches should help in evaluating cultivar adaptation and selecting cultivars with wide adaptation to environmental conditions [36,37]. When cultivars are evaluated across several years, the assessment is more reliable and cultivar means are characterized by a smaller error of estimation. Regardless of

whether we use controlled experiments, such as METs, or rely on farmer-survey data, the reliability of cultivar assessment increases with increasing numbers of years of evaluation. Assessments based on data from a single growing season are likely to be unreliable.

Due to their disadvantages and low credibility, yield data from farmer surveys should certainly not replace the METs data for cultivar recommendations. Survey yield data should only serve as complementary information to the classic cultivar recommendation. The use of survey data and the proposed statistical methodology of yield assessment allowed validation and suggested additional studies for improving the accuracy of cultivar recommendations and reducing the yield gap between MET and on-farm performances.

**Author Contributions:** Conceptualization, M.S., M.S.K. and W.M.; methodology, M.S., M.I. and T.O.; formal analysis, M.S. and E.W.-G.; data curation, M.I. and T.O.; writing—original draft preparation, M.S.; writing—review and editing, M.S.K., W.M. and E.W.-G.

**Conflicts of Interest:** The authors declare no conflict of interest.

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
