# Peer review of "Consistency of Yield Ranking and Adaptability Patterns of Winter Wheat Cultivars between Multi-Environmental Trials and Farmer Surveys"

_agronomy, doi:10.3390/agronomy9050245_

Round 1
Reviewer 1 Report
The manuscript entitlled "Evaluation of consistency of the yield ranking and adaptability patterns of winter wheat cultivars between multi-environmental trials and farmers' surveys" covers a quite interesting topic comparing the consistency between varietal choid derived from farmers' survey and multi location field trials in Poland. The topic is worth of investigation, honestly I found the discussion section a little bit confused since some variety names are often reported for the first time and do not clearly explaining why? for example why the cultivar Ozon is mentioned in the text and others are not? I suggest the authors to clarify the discussion, maybe with the use of additional graphs.
Specific comments:
please put at apex L57 the ha-1
in L 104 why the authors decided to use that code instead than more widespread BBCH?
when citing in the text specific reference authors should not report the authors' name and the year but the number of the reference under brackets. there are few citations that need to be re-formatted.
I suggest to improve the readability of figure 2 and 3, improving figure size, font size and maybe using a more easily understanable waty to present data, e.g. fig. 3 is not easy to read and the numbers near each cultivar are not explained anywhere.
Author Response
Dear Reviewer,
Thank you for your comments concerning our manuscript entitled "Evaluation of consistency of the yield ranking and adaptability patterns of winter wheat cultivars between multi-environmental trials and farmers' surveys". Those comments are all valuable and very helpful for revising and improving our paper. We have studied comments carefully and have made correction which we hope meet with approval. Below we present our answers to reviewercomments
The manuscript entitlled "Evaluation of consistency of the yield ranking and adaptability patterns of winter wheat cultivars between multi-environmental trials and farmers' surveys" covers a quite interesting topic comparing the consistency between varietal choid derived from farmers' survey and multi location field trials in Poland.
The topic is worth of investigation, honestly I found the discussion section a little bit confused since some variety names are often reported for the first time and do not clearly explaining why? for example why the cultivar Ozon is mentioned in the text and others are not? I suggest the authors to clarify the discussion, maybe with the use of additional graphs.
Response: We accept this remarks, in revised manuscript in additional obtain, which will allow on better assessment of the cultivars yield adaptability. According to this suggestion and new analysis we supplemented the Discussion section. We hope that now the discussion is less confusing and more unambiguous.
Specific comments:
please put at apex L57 the ha-1
Response: It was done.
in L 104 why the authors decided to use that code instead than more widespread BBCH?
Response: We accepted this remarks, in revised manuscript we used the BBCH scale. In fact, BBCH and Zadoks scale are similar for cereals.
when citing in the text specific reference authors should not report the authors' name and the year but the number of the reference under brackets. there are few citations that need to be re-formatted.
Response: In revised manuscript the some citations were corrected.
I suggest to improve the readability of figure 2 and 3, improving figure size, font size and maybe using a more easily understanable waty to present data, e.g. fig. 3 is not easy to read and the numbers near each cultivar are not explained anywhere.
Response: We accepted this remarks, in revised manuscript we prepare new version of Figure 2. However, in new version we resigned from Figure 3, according to suggestion Reviewer 2.
Reviewer 2 Report
agronomy-465060
2019-04-01
This is an interesting study and the paper is well presented. However, there are important errors in data interpretation. The most important results were presented in Fig. 2. From this figure the following conclusions can be arrived:
The six regions were quite similar in terms of cultivar ranking, particularly for the survey data. This indicate that the separation of the wheat growing area into 6 regions is questionable, and tit is probably better not to analyze by region. A joint analysis across all regions would be more valid
Based on performance across regions, KWS_Ozon was clearly the best for Survey data, MIM, and HIM.
Using the mean-vs-stability form of the GGE biplot would be more effective in seeing the rank of the cultivars in Survey, MIM, and HIM, as well as the similarity/dissimilarity among them. I expect the conclusion would be different from what they see in Table 2.
Figure 3 is misleading and should be not used. From this figure the cultivar Bomberka was identified as the most stable cultivar. However, it was one of the poorest yielding cultivars, according to Fig. 2. mean-vs-stability form of the GGE biplot would allow visualization of both high yielding and stability across regions for Survey data, MIM, and HIM.
The equations should use equation editors
There are some grammar errors but I did not record them.
Author Response
Dear Reviewer,
Thank you for your comments concerning our manuscript entitled "Evaluation of consistency of the yield ranking and adaptability patterns of winter wheat cultivars between multi-environmental trials and farmers' surveys". Those comments are all valuable and very helpful for revising and improving our paper. We have studied comments carefully and have made correction which we hope meet with approval. Below we present our answers to reviewer comments
This is an interesting study and the paper is well presented. However, there are important errors in data interpretation. The most important results were presented in Fig. 2. From this figure the following conclusions can be arrived:
The six regions were quite similar in terms of cultivar ranking, particularly for the survey data. This indicate that the separation of the wheat growing area into 6 regions is questionable, and tit is probably better not to analyze by region. A joint analysis across all regions would be more valid
Response: In revised manuscript we have expanded our analysis to determine the value of superiority measure Pi obtained across study regions. In return for assessing the cultivars stability presented in Figure 3. Unfortunately, the division into these 6 regions is valid in this system for the cultivars evaluation and recommendation for farmers. One of our goals was to determine whether it is appropriate and allows a reliable recommendation of cultivars. As one of our proposals, we suggest dividing Poland into different regions.
Based on performance across regions, KWS_Ozon was clearly the best for Survey data, MIM, and HIM.
Using the mean-vs-stability form of the GGE biplot would be more effective in seeing the rank of the cultivars in Survey, MIM, and HIM, as well as the similarity/dissimilarity among them. I expect the conclusion would be different from what they see in Table 2.
Response: In revised manuscript we have expanded our analysis to determine the value of superiority measure Pi. This index obtained across regions provides the similar results and conclusions as the mean-vs-stability GGE biplot. Therefore, we have not added this type of GGE biplot to manuscript.
Figure 3 is misleading and should be not used. From this figure the cultivar Bomberka was identified as the most stable cultivar. However, it was one of the poorest yielding cultivars, according to Fig. 2. mean-vs-stability form of the GGE biplot would allow visualization of both high yielding and stability across regions for Survey data, MIM, and HIM.
Response: We accepted this remarks, in revised manuscript we resigned from Figure 3.
The equations should use equation editors
Response: It was done.
There are some grammar errors but I did not record them.
Response: We edited manuscript according to this suggestion.
Reviewer 3 Report
I apologise for not submitting this report on Sunday as promised.
However, the wifi in this hotel where I am located in China was not
operating until yesterday. Now I cannot download the report form.
The
paper is average in content and presentation. It is verbose in places
and needs considerable modification to the English presentation. The
referencing in the test is a mixture of name and numbers and the actual
reference list often includes uppercase first letters. The authors
should be encouraged to pay more attention to journal requirements.
A tracked copy of the manuscript with suggestions is attached.
The idea of on-farm trialing is not new. Many plant breeding companies
perform advanced trials on farm with varying inputs of management by the
farmers and farm machinery. This provides the kind of data discussed in
the present manuscript except that the data is collected from farmer
surveys rather than actual on-farm trials by scientists.

Author Response
Dear Reviewer,
Thank you for your comments concerning our manuscript entitled "Evaluation of consistency of the yield ranking and adaptability patterns of winter wheat cultivars between multi-environmental trials and farmers' surveys". Those comments are all valuable and very helpful for revising and improving our paper. We have studied comments carefully and have made correction which we hope meet with approval. Below we present our answers to reviewer comments
Reviewier 3
The paper is average in content and presentation. It is verbose in places and needs considerable modification to the English presentation. The referencing in the test is a mixture of name and numbers and the actual reference list often includes uppercase first letters. The authors should be encouraged to pay more attention to journal requirements.
Response: The revised version of the manuscript was edited according to this suggestions.
A tracked copy of the manuscript with suggestions is attached.
Response: All suggestions in the copy of manuscript have been considered and accepted
The idea of on-farm trialing is not new. Many plant breeding companies perform advanced trials on farm with varying inputs of management by the farmers and farm machinery. This provides the kind of data discussed in the present manuscript except that the data is collected from farmer surveys rather than actual on-farm trials by scientists.
Round 2
Reviewer 2 Report
L739: “In wEMPCA, for a more accurate estimation of adjusted means, the weights should be greater when adjusted means are estimated with larger errors.” The weights were the inverse of the errors!
L827-838: in your response you have rejected the reviewer’s comment that the eco-regional were not justified by your data and stated that the regional were valid without any justification. If it is valid, why do you want evaluate genotypes across all regions?!
As an aside, your Pi should be highly correlated with the mean yield (i.e., cultivar main effects). Please check it out. If Pi is highly correlated with the mean yield, would you still want to use it?
It looks like you believe surrey results are more reliable than yield trials. Is this true, and why? If this belief is justified, then your GGE biplot for the survey data strongly indicates that the yield rank of the cultivars were closely correlated between “regions” and the same best cultivars were identified for all regions. Is this true? If this is true, why do you still insist separate analyses by regions and try to interpret the poor consistency in some regions? You actually did not good interpretations. One possible reason is that the number of test locations were too few to represent the region for reliable cultivar evaluation. Increase the number of test locations, or use locations from other regions (this is justified as the regions are positively correlated) may improve cultivar evaluation.
You conclusions were contradictory to each other and will be difficult for farmers to accept. On one hand, you discussed the poor consistency between survey results in HIM system by region. Then in another paragraph, you discussed one cultivar was the best for both the survey data and the HIM.
Another contradiction is your view of the reliability of the survey data. On one hand, when you see a poor consistency, you blame yield trials not representing the region. On the other, you discussed it is difficult to obtain reliable survey data. Such contradictions will only confuse your intended readers. A safer approach is to say that both survey data and yield trial data have their merits and limitations and results from both should be used complementarily in recommending cultivars, to subregions or to the whole Poland.
Author Response
Dear Reviewer,
Below we present our answers to reviewer comments.
L739: “In wEMPCA, for a more accurate estimation of adjusted means, the weights should be greater when adjusted means are estimated with larger errors.” The weights were the inverse of the errors!
Response: You're correct. We modified this sentence in the revised manuscript.
L827-838: in your response you have rejected the reviewer’s comment that the eco-regional were not justified by your data and stated that the regional were valid without any justification. If it is valid, why do you want evaluate genotypes across all regions?!
Response: Unfortunately, the method of collecting data in our study prevents us from comparing MET data differently than the proposed division of growing environments into regions. We cannot divide yield data from surveys into other agro-ecological regions or individual locations. That's why we could not take into account the previous reviewer's comments. Your comment is appreciated, but unfortunately the nature of the data did not allow us to take it into account.
As an aside, your Pi should be highly correlated with the mean yield (i.e., cultivar main effects). Please check it out. If Pi is highly correlated with the mean yield, would you still want to use it?
Response: Indeed, there is a strong correlation here. In the new version of manuscript, we have dropped this analysis.
It looks like you believe surrey results are more reliable than yield trials. Is this true, and why? If this belief is justified, then your GGE biplot for the survey data strongly indicates that the yield rank of the cultivars were closely correlated between “regions” and the same best cultivars were identified for all regions. Is this true? If this is true, why do you still insist separate analyses by regions and try to interpret the poor consistency in some regions? You actually did not good interpretations. One possible reason is that the number of test locations were too few to represent the region for reliable cultivar evaluation. Increase the number of test locations, or use locations from other regions (this is justified as the regions are positively correlated) may improve cultivar evaluation.
You conclusions were contradictory to each other and will be difficult for farmers to accept. On one hand, you discussed the poor consistency between survey results in HIM system by region. Then in another paragraph, you discussed one cultivar was the best for both the survey data and the HIM.
Another contradiction is your view of the reliability of the survey data. On one hand, when you see a poor consistency, you blame yield trials not representing the region. On the other, you discussed it is difficult to obtain reliable survey data. Such contradictions will only confuse your intended readers. A safer approach is to say that both survey data and yield trial data have their merits and limitations and results from both should be used complementarily in recommending cultivars, to subregions or to the whole Poland.
Response: We will answer the above comments together. The previous version of the discussion could have suggested that the survey data and analysis based on them were more reliable. As the reviewer rightly notes, this is not true. In the discussion in the revised version, we have deleted the sentences and terms that would suggest that the survey data are better. We supplemented the discussions about the possible reasons for the low credibility of the farmer surveys. In the discussion, we also deleted the indication of specific cultivars as widely adapted. The purpose of our study is not to make cultivar recommendation and indicate cultivars with wide adaptation. The purpose was only to present the possibility of using survey yield data to improve recommendations. The Discussion section has been improved in accordance with the reviewer suggestions.
Reviewer 3 Report
A few minor suggestions might be considered: Referring to the tracked PDF:
Line 21: Italicise the species name
line 35: '...labeled as widely...'
Line 218: Maybe the reference citation should be [7,10,16]
Line 219: 'Low yields might result from the use of........'
Line 22: Change 'effective' to 'reliable'
Line 534: '....dataset is:' Delete 'given below'
Line 742: '...using the formula:'
Line 828: '....according to:'
Line 949 '.....level for the MET....'
Line 1042: '' In the survey data sets this factor...'
Line 1362: Another difficulty in applying....'
Line 1368: '....experiments with similar numbers of.....'
Line 1375:' especially the tools....'
Author Response
Dear Reviewer,
Below we present our answers to reviewer comments
A few minor suggestions might be considered: Referring to the tracked PDF:
Line 21: Italicise the species name
line 35: '...labeled as widely...'
Line 218: Maybe the reference citation should be [7,10,16]
Line 219: 'Low yields might result from the use of........'
Line 22: Change 'effective' to 'reliable'
Line 534: '....dataset is:' Delete 'given below'
Line 742: '...using the formula:'
Line 828: '....according to:'
Line 949 '.....level for the MET....'
Line 1042: '' In the survey data sets this factor...'
Line 1362: Another difficulty in applying....'
Line 1368: '....experiments with similar numbers of.....'
Line 1375:' especially the tools....'
Response: All of the above comments have been incorporated in the revised version of manuscript.